# Effectiveness of the capability approach in rehabilitation for persons with neuromuscular diseases: A controlled before-after study

Eirlys J. Pijpers[1]*, Bart Bloemen[2], Wija J. Oortwijn[3], Baziel G. M. van Engelen[4], Gert Jan van der Wilt[2], Jan T. Groothuis[1], Edith H. C. Cup[1]

**1** Radboud university medical center, Donders Institute for Brain, Cognition and Behaviour, Department of Rehabilitation, Nijmegen, The Netherlands, **2** Radboud university medical center, Donders Institute for Brain, Cognition and Behaviour, IQ Health science Department, Nijmegen, The Netherlands, **3** Radboud university medical center, Research Institute for Medical Innovation, IQ Health science Department, Nijmegen, The Netherlands, **4** Radboud university medical center, Donders Institute for Brain, Cognition and Behaviour, Department of Neurology, Nijmegen, The Netherlands

* Eirlys.pijpers@radboudumc.nl

## Abstract

### Background

Rehabilitation of persons with neuromuscular diseases (NMD) requires a personalised approach that focuses on achieving and maintaining a level of functioning that enables them to be in a state of well-being. The capability approach states that well-being should be understood in terms of capabilities, which are the opportunities that people have to be and do things they have reason to value. The aim of this study is to investigate whether providing care inspired by the capability approach (capability care) has an effect on the well-being of persons with NMD.

### Methods

In the Rehabilitation and Capability care for persons with NeuroMuscular Diseases (ReCap-NMD) study, 64 adults with facioscapulohumeral muscular dystrophy or myotonic dystrophy type 1 were included in two groups, using a before-after controlled design with 6 months between the measurement moments. The first group received rehabilitation as usual, the second received capability care. This article reports on the primary outcome measure, the Canadian Occupational Performance Measure (COPM) and secondary quantitative outcome measures (questionnaires on participation, capability well-being and health-related quality of life).

### Results

There was no difference between capability care and usual care on the COPM and the secondary outcome measures. There was a similar improvement for both

**Data availability statement:** The data generated and analyzed for this study is stored in a Data Acquisition Collection (DAC) in the Radboud Data Repository (https://doi.org/10.34973/c7cr-0z09) to which access can only be obtained after filing a request through the Radboud Data Repository. To request access, one needs to make an account for the Radboud Data Repository and request access through this system. The information on these steps can be found on this webpage: https://data.ru.nl/doc/help/helppages/visitor-manual/vm-request-access.html.

**Funding:** This work was supported by the Prinses Beatrix Spierfonds (https://www.spierfonds.nl/). BvE, GvdW, JG and EC received this funding under grant W.OK18-08. The funder had no role in the study design, writing of the paper or decision to submit for publication.

**Competing interests:** I have read the journal's policy and the authors of this manuscript have the following competing interests: Jan Groothuis currently serves as an academic editor for PLOS ONE. He is also a member of the Fulcrum therapeutics European medical advisory board (honorarium to employer). All other authors have declared that no competing interests exist.

capability care and usual rehabilitation on the COPM at 6-month follow-up. This means that the effect of capability care is similar to usual rehabilitation.

## Conclusion

This is the first study that explicitly developed, implemented and evaluated a clinical healthcare intervention inspired by the capability approach. We found no difference on the COPM between persons with NMD receiving capability care compared to usual rehabilitation. There is a need for further research on the clinical relevance and added value of the capability approach for rehabilitation and other fields in healthcare.

## Trial registration

Trialregister.nl NL8946.

## Introduction

For persons with neuromuscular diseases (NMD), where no curative treatment is available, rehabilitation focuses on achieving and maintaining a level of functioning that enables them to lead a satisfying life. Myotonic dystrophy type 1 (DM1) and facioscapulohumeral muscular dystrophy (FSHD) are among the most prevalent inherited NMD [1,2]. Both diseases are characterized by slowly progressive muscle weakness leading to impairments in functions, limitations in activities and participation restrictions [3,4]. In addition, DM1 is a multisystem disorder with cardiac and gastrointestinal involvement, cataract, and cognitive deficits [3,5]. Both diseases show a highly variable disease course and severity among patients. Therefore, rehabilitation treatment requires an individualized approach for maintaining or improving their level of functioning, leading to a higher well-being.

With regard to well-being, within healthcare there is a growing interest in the use of the capability approach. It offers a theoretical framework for studying well-being by analysing and evaluating an individual's ability to achieve valuable *functionings* in life [6–8]. Its central idea is that well-being is a result of a person's *capabilities*, the set of opportunities available that a person can choose from to realise valuable *functionings*. These are defined as the beings and doings that people have reason to value (e.g., working, resting, sports and leisure activities, being part of a family, belonging to a community). Person's capabilities are determined by the *resources* at someone's disposal and personal, social and environmental characteristics (*conversion factors*). The larger the number of capabilities, the more options a person has to choose from to realise valuable functionings and the higher this person's well-being will be [6,7,9].

Using the capability approach may add value to making the shift towards a broader perspective on rehabilitation with an explicit focus on well-being. Thereby, it also offers a direction for analysis of the factors (resources, conversion factors) that impact on a person's well-being. Until now, the capability approach has not yet been

used for clinical intervention development. In the healthcare field it has been primarily used for evaluating the impact of health-related states and interventions on well-being [10,11].

The Rehabilitation and Capability care in NMD (ReCap-NMD) study [12] explicitly developed, implemented and evaluated a clinical healthcare intervention inspired by the capability approach. We hypothesised that implementing the capability approach in rehabilitation for persons with slowly progressive NMD might broaden the focus that is needed to fully provide person-centred care, potentially leading to a higher well-being for these persons. The development of *capability care, i.e.,* rehabilitation inspired by the capability approach, is described elsewhere [13]. The aim of this article is to present the study results regarding the effectiveness of capability care compared to usual rehabilitation care in persons with NMD.

## Materials and methods

### Study design

Two groups of participants were included in a controlled before and after design, using a mixed-methods approach to assess the effect of rehabilitation on their well-being. The first group (of 30 participants) received rehabilitation care as usual. Next, the healthcare professionals were trained in providing the developed *capability care.* Then, the second group (of 30 participants) received *capability care.* Participants were blinded for treatment group. Data were collected at baseline (T0) and 6 months after the initial measurements (follow-up, T1). After the T1 measurements participants were asked whether they had a suspicion of treatment group and if they could elaborate on this suspicion. A more extensive description of the study design [12] and the *capability care* intervention [13] are published separately.

### Participants

Participants were recruited at the Radboud university medical center of expertise for NMD, Nijmegen, The Netherlands, via the departments of Rehabilitation and Neurology between 11 November 2020 and 21 June 2023. All persons with planned appointments were screened by their rehabilitation physician or nurse practitioner for eligibility. Participants were recruited and included before their yearly appointments for 'analysis and advice' at the expertise outpatient rehabilitation clinic of the department of Rehabilitation. The study protocol was approved by the medical research ethics committee CMO Regio Arnhem-Nijmegen (NL72794.091.20) and registered at trialregister.nl (NL8946) on 12 October 2020. All participants provided written informed consent. Participants were adults with a diagnosis of FSHD or DM1 with a current rehabilitation aim. See Table 1 for an overview of all inclusion and exclusion criteria.

### Intervention

Participants in both groups had one day of appointments for 'analysis and advice' at the multidisciplinary outpatient rehabilitation clinic. As the Radboud university medical center is a Center of Expertise for both FSHD and DM1, persons from throughout the Netherlands have appointments with the multidisciplinary rehabilitation team. Prior to their visit, it was

**Table 1. Inclusion and exclusion criteria of the ReCap-NMD study.**

| Inclusion criteria | Exclusion criteria |
|---|---|
| 1) confirmed genetically proven diagnosis of FSHD or DM1<br>2) 18 years or older<br>3) a current rehabilitation aim<br>4) in a mentally stable condition<br>5) sufficient mastery of the Dutch language to participate in conversation with the healthcare providers and research assistant and to fill in questionnaires | 1) active or previously major psychotic, psychiatric or depression episodes<br>2) acquired brain injury (e.g., stroke, traumatic brain injury)<br>3) severe cognitive problems (e.g., severe dementia) in which case the rehabilitation treatment is affected and/or patients are not able to fill out the questionnaires<br>4) limited life expectancy (e.g., due to cancer) |

determined which healthcare professionals they have consultations with during this day. This was determined together with the person with NMD and depended on their type of NMD and personal circumstances. They always had an appointment with the rehabilitation physician and with one, several or all of the following healthcare professionals: nurse practitioner, physical therapist, occupational therapist, speech and language therapist and/or dietician. After the individual consultations, the multidisciplinary team discussed their findings to provide a rehabilitation advice, which was communicated and discussed with the person with NMD at an additional (phone) appointment with the rehabilitation physician, to make a shared decision on the next steps.

The usual care group received rehabilitation care as usual at the outpatient rehabilitation clinic. The capability care group received *capability care:* the multidisciplinary team was trained to provide care from a capability perspective. The structure of the day of appointments is similar for both groups. What is changed in capability care is the focus of the conversations between the person with NMD and the healthcare professionals, as well as the analysis and advice by the multidisciplinary rehabilitation team, which were conducted, discussed and formulated from a capability perspective. *Capability care* can focus both on facilitating and achieving *functionings* (beings and doings) as well as looking for alternative *functionings* that fulfil the same underlying value, thereby contributing to a persons' well-being. To facilitate a conversation on broader aspects that impact on well-being, persons with NMD receive a preparation letter and healthcare professionals are provided with guiding questions and practical tools to use. A more extensive description of capability care and its development is published elsewhere [13].

## Outcome measures

We selected the Canadian Occupational Performance Measure performance (COPM-P) and satisfaction (COPM-S) scales as primary outcome measure to assess the effect of capability care and usual rehabilitation on participants well-being. According to the capability approach, well-being is determined by the capabilities a person has. As there is no measure to determine all possible capabilities, we chose to measure realised capabilities; these are the beings and doings (*functionings* in the capability approach). The COPM is an individualized outcome measure that identifies priorities in a person's meaningful occupations. We therefore used this as a proxy for measuring a change in functionings, implying that a change in capabilities has also taken place.

The primary outcome was the mean score of the participants self-rated performance (COPM-P) on, and satisfaction (COPM-S) with 3–5 self-selected daily occupations [14]. Through a semi-structured interview, participants identified and prioritized 3–5 problems in their occupational performance. These occupations were subsequently rated on a 10-point scale for perceived performance (1 = impossible; 10 = very well able) and similar for perceived satisfaction (1 = not satisfied at all; 10 = fully satisfied). The COPM is a reliable and valid instrument [15–17], and has been shown to be a sensitive and clinically relevant outcome measure in a number of rehabilitation studies [15,18–21].

To uncover the diverse effects of care, we selected a broad range of secondary outcome measures on participation, capability well-being, and health-related quality of life. These included the Dutch versions of the Utrecht Scale for Evaluation of Rehabilitation Participation (USER-P) [22], EuroQol-5D-5L (EQ-5D-5L) [23,24], ICEpop CAPability measure for Adults (ICECAP-A)[25,26] and Medical Outcome Study Short Form-36 v2 (SF-36) [27]. For the USER-P, we used the scores on the three subscales frequency, restrictions, and satisfaction. These scores have a range from 0–100 with higher scores indicating higher participation (higher frequency, less restrictions, more satisfaction) [22,28]. For the EQ-5D-5L we used the visual analogue scale (VAS) and the level sum score (LSS). The VAS ranges from 0–100 with a higher score indicating a better health; the LSS ranges from 5–25 with a higher score indicating a worse health state [29,30]. The ICECAP-A comprises 5 items, which are capabilities important to one's well-being: (1) stability, i.e., being able to feel settled and secure in all areas of life; (2) attachment, i.e., being able to have love, friendship, and support; (3) autonomy, i.e., being able to be independent; (4) achievement, i.e., being able to achieve and progress in all aspects of life; (5) enjoyment, i.e., being able to have a lot of enjoyment and pleasure [25]. We used the sum score ranging from 4–20 and the

tariff value (calculated based on the general UK population [31]) ranging from 0–1. Higher scores indicate a higher capability well-being [25]. The SF-36 consists of eight subscales (physical functioning, role limitations due to physical health problems, role limitations due to emotional health problems, vitality, mental health, social functioning, bodily pain, general health) with scores ranging from 0–100; a higher score indicating a higher health-related quality of life [27,32].

## Sample size calculation

The sample size was calculated based on the primary outcome measure. The expected difference between both groups is 1.4 for COPM-P and 1.9 for COPM-S. These values represent a clinically relevant improvement as perceived by the person [19]. The standard deviations in both groups and for both COPM-P and COPM-S are expected to be 1.9 at 6 months follow-up (T1) [20]. Based on these assumptions, a sample size of at least 24 participants per group will yield a power of 80% to show a statistical difference between usual rehabilitation and capability care group at a significance level of 0.05. These calculations are based on ANCOVA-models with the outcomes at 6 months as dependent variables, corrected for the value at baseline (T0) and an assumed correlation of 0.5 between the outcome values at T0 and T1 [33]. Considering dropout, we aimed to include 30 participants in each group. We aimed for 20 participants with FSHD and 10 with DM1 per group, as more persons with FSHD visit the outpatient clinic than persons with DM1.

## Data collection

Participant data were collected 1–2 weeks before their appointments at the department of Rehabilitation (baseline, T0) and 6 months after the initial measurements (follow-up, T1). The primary outcome measure, the COPM, is a semi-structured interview and was administered by an independent research assistant (occupational therapist) during an online or phone appointment. The secondary outcome measures were questionnaires sent to the participants using the electronic data platform Castor EDC [34]. The results of the COPM were manually entered into Castor EDC by the research assistant. The COPM results are also reported in the electronic health record visible to the healthcare team (so that the occupational therapist does not need to repeat the COPM, as it is often used in clinical practice). Adverse events or irregularities affecting protocol adherence were registered in Castor EDC by the primary researchers (BB and EP).

## Statistical analysis

As described in the study protocol [12], the primary outcome measure was analysed using ANCOVA models with the follow-up scores at T1 of both groups as dependent variable. Covariates were the scores at baseline (T0) of both groups, age, sex and type of NMD. Group was a fixed factor. For an intervention effect to be present, both the COPM-P and COPM-S needed to show a significant difference between groups ($p < 0.05$).

To determine whether both capability care and usual rehabilitation had an effect, paired T-tests were performed for the capability care and usual care group separately. The dependent variables were the T0 and T1 scores on the COPM-P and COPM-S. For an intervention effect to be present, both the COPM-P and COPM-S needed to show a significant effect (one-sided $p < 0.025$); we hypothesized an improvement on T1 compared to T0.

For the secondary outcome measures, ANCOVA models were fitted with the follow-up scores at T1 as dependent variables. Covariates were the T0 score of both groups, age, sex and type of NMD. Group was a fixed factor. Additionally, paired T-tests were performed for the secondary outcome measures, similar to those described for the primary outcome measure.

Incomplete measures (i.e., a participant that did not complete all COPM interviews and/or questionnaires) were excluded from analysis. If only part of the data from a subject was missing, this subject was not excluded from all analyses. For example, if the primary outcome measure was complete but the secondary outcome measures were missing, this subject was only excluded for analysis of the secondary outcome measures. All statistical analyses were performed using Statistical Package for Social Sciences (SPSS) version 29.

## Results

### Participants

A total of 101 persons were pre-screened by their physician or nurse practitioner and contacted by a researcher (EP or BB). Of these, 64 persons agreed to participate (see Fig 1). The foremost reasons for declining participation (similar for both types of NMD) were: rehabilitation appointments planned too short to be able to complete baseline measurements in time (n = 10), too time or energy consuming (n = 8), inclusion already complete for their type of NMD (FSHD or DM1) (n = 6) (pre-screening had continued, while sufficient participants with the type of NMD were already included) and no current rehabilitation aims (n = 5) (see Fig 1). Of the 64 participants, six were excluded or dropped out during the study. Three (two DM1, one FSHD) could not formulate a current rehabilitation aim during the COPM interview with the research assistant, one postponed appointments to a time where capability care training of the professionals had already started (and therefore could no longer be included in the usual care group), one cancelled appointments, and we lost contact with one before the first measurement. In total, 58 participants completed baseline measurements for the primary outcome measure; however, one participant did not complete any of the T1 measurements and was excluded from analysis. Therefore,

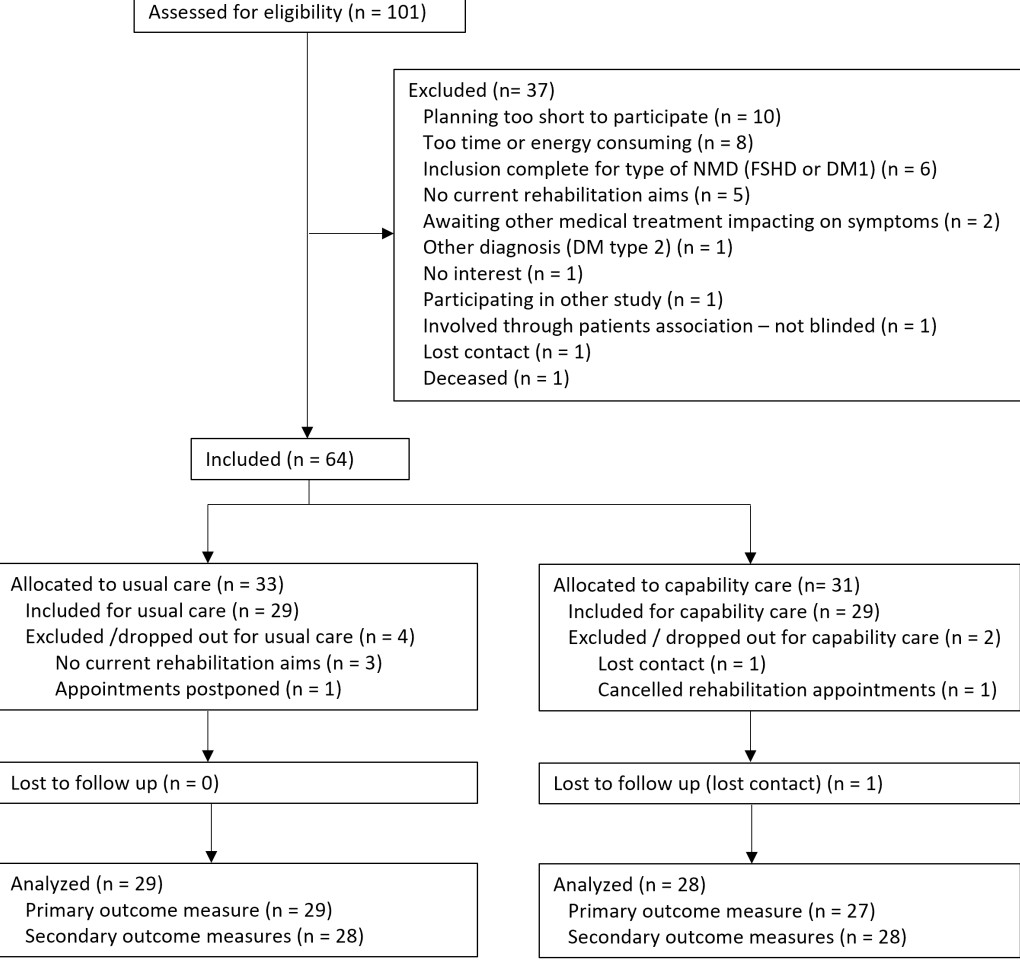

**Fig 1. Flowchart of the inclusion process.** DM1: Myotonic dystrophy type 1; FSHD: Facioscapulohumeral muscular dystrophy; NMD: neuromuscular disease.

data from 57 participants were used for analysis. Of these, three participants did not complete all measurements (see Fig 1 and Table 2). Additionally, one participant only completed the ICECAP-A and no other secondary outcome measures.

For the usual care group, baseline measurements took place between November 2020 and August 2021, with follow-up between May 2021 and February 2022. For the capability care group, baseline measurements took place between March 2022 and December 2022 with follow-up between October 2022 and June 2023.

Table 2 shows the characteristics of the participants included for analysis for both usual and capability care. The groups seemed comparable at first glance with respect to type of NMD, sex, age (Table 2) and baseline values of the primary and secondary outcome measures (Table 5 and S1-S5 Tables).

Table 3 shows whether participants had a suspicion of which treatment group they were assigned to. In the usual care group, 28% had a correct suspicion. In the capability care group, 43% had a correct suspicion. When we look at the reasons that participants provided, in each group 4 participants that suspected usual care mentioned that the care is not different from what they had experienced previously. In the usual care group all 7 participants (24%) that suspected capability care provided reasons such as personalised treatment, professionals took the time for my story and they really helped me with my questions; in the capability care group 3 participants (11%) provided similar reasons. In the capability care group 6 participants (21%) provided reasons that could be attributed to capability care, such as a focus on my possibilities (1 participant), attention for mental health as well as physical symptoms (1 participant), a different way of asking questions (1 participant) and attention for values (3 participants).

## Canadian Occupational Performance Measure

We fitted ANCOVA models to determine whether capability care had a larger effect on the COPM-P and COPM-S scales compared to usual rehabilitation. The covariates age, sex and type of NMD had no significant effect on the outcome.

**Table 2. Baseline characteristics of the participants included for analysis.**

| | | Usual care | Missing data | | Capability care | Missing data | |
| --- | --- | --- | --- | --- | --- | --- | --- |
| | | | Primary outcome measure | Secondary outcome measure | | Primary outcome measure | Secondary outcome measure |
| Total N | | 29 | 0 | 2 | 28 | 1 | 0 |
| Type of NMD | FSHD | 19 (66%) | | 2 | 18 (64%) | | |
| | DM1 | 10 (34%) | | | 10 (36%) | 1 | |
| Sex | Male | 11 (38%) | | 1 | 14 (50%) | 1 | |
| | Female | 18 (62%) | | 1 | 14 (50%) | | |
| Age | Mean (Range, SD) | 50.8 (24-71, 12.8) | | | 47.2 (30-68, 10.1) | | |

DM1: Myotonic dystrophy type 1; FSHD: Facioscapulohumeral muscular dystrophy; NMD: neuromuscular disease; SD: standard deviation.

**Table 3. Suspicion of treatment group by participants.**

| | Usual care (n = 29) | Capability care (n = 28) |
| --- | --- | --- |
| No suspicion | 14 (48%) | 7 (25%) |
| Suspects usual care | 8 (28%) | 6 (21%) |
| Suspects capability care | 7 (24%) | 12 (43%) |
| Missing data | 0 | 3 (11%) |

The baseline score had a significant effect on the outcome for both COPM-P (F(1,50) = 25.852, p<0.001) and COPM-S (F(1,50) = 13,840, p<0.001). Group showed no significant effect for neither COPM-P (F(1,50) = 0.004, p=0.947) or COPM-S (F(1,50) = 0.401, p=0.530) (see also Table 4).

Paired T-tests were performed to determine whether there was a difference between baseline and follow-up for both groups. Means and standard deviations for both groups are shown in Table 5. At follow-up, both the usual care and capability care group showed an improvement on COPM-P (t(28) = 4.104, p<0.001 and t(26) = 2.995, p=0.003, respectively) and COPM-S (t(28) = 4.564, p<0.001 and t(26) = 3.433, p=0.001, respectively).

## Secondary outcome measures

ANCOVA models were fitted to determine whether capability care had a larger effect than usual rehabilitation care on all secondary measures. An overview of the baseline and follow-up scores of the secondary measures can be found in the supporting information (S1-S5 Tables). For all measures, the baseline score of the respective subscales showed a significant effect (all p<0.001) on the outcome. For 6 of the 15 subscales, some of the covariates age, sex and type of NMD showed a significant effect (see S5 Tables). Group showed no significant effect for all subscales of all secondary outcome measures.

Paired T-tests were performed to determine whether there was a difference between baseline and follow-up for both groups. These showed no significant difference between baseline and follow-up for all secondary outcome measures.

## Discussion

To the best of our knowledge this is the first study to explicitly develop, implement and evaluate a clinical healthcare intervention inspired by the capability approach. Our results showed no difference between capability care and usual care on the COPM and a range of questionnaires on participation (USER-P), capability well-being (ICECAP-A) and health-related

**Table 4. Confidence intervals and p-values for the effect of group on the Canadian Occupational Performance Measure performance (COPM-P) and satisfaction (COPM-S) scales.**

|  | Mean difference | 95% CI for difference | | P-value |
|---|---|---|---|---|
|  |  | Lower | Upper |  |
| COPM-P | 0.2 | −0.5 | 0.6 | 0.947 |
| COPM-S | 0.2 | −0.5 | 0.9 | 0.530 |

CI: confidence interval; COPM-P: Canadian Occupational Performance Measure Performance score; COPM-S: Canadian Occupational Performance Measure Satisfaction Score.

**Table 5. Means and standard deviations of the scores on the Canadian Occupational Performance Measure performance (COPM-P) and satisfaction (COPM-S) scales for both groups at baseline (T0) and 6-month follow-up (T1).**

|  |  | Baseline (T0) | | 6-month follow-up (T1) | | Mean difference | 95% CI of the difference | | P-value (one sided) |
|---|---|---|---|---|---|---|---|---|---|
|  |  | Mean | SD | Mean | SD |  | Lower | Upper |  |
| COPM-P | Usual care (n=29) | 4.6 | 1.5 | 5.5 | 1.3 | 1.0 | 0.5 | 1.4 | < 0.001 |
|  | Capability care (n=27) | 5.0 | 1.7 | 5.8 | 1.1 | 0.8 | 0.2 | 1.3 | 0.003 |
| COPM-S | Usual care (n=29) | 4.4 | 1.6 | 5.5 | 1.3 | 1.1 | 0.6 | 1.6 | < 0.001 |
|  | Capability care (n=27) | 4.7 | 1.4 | 5.8 | 1.3 | 1.1 | 0.4 | 1.7 | 0.001 |

CI: confidence interval; COPM-P: Canadian Occupational Performance Measure Performance score; COPM-S: Canadian Occupational Performance Measure Satisfaction Score. One-sided P-values for the T-test are shown.

quality of life (EQ-5D-5L, SF-36). There were improvements at follow-up on the COPM scales compared to baseline for both groups. This suggests that the effects of capability care and usual rehabilitation in this group of persons with NMD were comparable. However, secondary outcome measures showed no difference between baseline and follow-up.

**Comparing capability care to usual care**

Overall, we found no significant differences between the capability care group and usual care group. This might indicate insufficient contrast between the two groups, which may be due to: 1) the choice of outcome measures; 2) the content of capability care; and/or 3) the implementation of capability care. Below, we will discuss each of these options.

**Outcome measures.** The aim of the study was to determine whether rehabilitation inspired by the capability approach had a larger effect on well-being of persons with NMD than usual rehabilitation. Well-being and quality of life are terms with a wide range of definitions and are sometimes used interchangeably [35–38]. The capability approach conceptualises well-being in terms of capabilities: the effective opportunities people have to realise what is of value to them (i.e., functionings) [6,7].

We used a range of outcome measures, with the COPM as primary outcome measure for measuring individualized occupations that were important for the person to improve. The COPM is a commonly used measure in both rehabilitation practice and research [14–17,19,39], has been used for evaluating interventions for persons with NMD [20] and has repeatedly shown to have satisfactory to excellent measurement properties (reliability, validity and responsiveness) [15]. We chose the COPM as daily occupations (the construct of the COPM) represent *functionings* in the capability approach; assuming that an improvement in capabilities would lead to an improvement on the functionings, represented by the occupations assessed with the COPM. Thus, we used the COPM as a proxy for measuring capability well-being. However, when developing capability care, we realised that an important difference with usual rehabilitation is the focus on broader aspects that impact on well-being. As the COPM focuses on daily occupations, it might not capture changes in all relevant capabilities or functionings, especially those capabilities that do not directly relate to occupations. For example, the dimension *life* which includes feeling comfortable and at ease with oneself; or *practical reasonableness* that includes being able to make plans and decisions.

One of our secondary outcome measures is the ICECAP-A, which measures capability well-being, and does include broader aspects that impact on well-being [25]. However, a limitation of the ICECAP-A is that it is not individualized. It does therefore not capture changes in capabilities important to that person specifically; whereas rehabilitation especially focuses on achieving individualized goals. A personalised measure might be needed to capture changes in these specific functionings that are of value to a person. A mixed methods analysis comparing the COPM, ICECAP-A and participant interviews has been performed (manuscript submitted) to provide more insight into the content validity and responsiveness of these measures in the context of rehabilitation for persons with NMD.

Furthermore, we included commonly used health-related quality of life (EQ-5D and SF-36) and participation (USER-P) outcome measures. These measures, however, only capture a part of well-being or quality of life and were included to uncover the diverse effects of care. Moreover, these generic outcome measures might be unresponsive in rare diseases such as FSHD and DM1 [40,41]. Furthermore, on the health-related quality of life measures, no change or even a deterioration might be expected due to progression of the NMD.

In addition, we did not include measures that included persons' expectations or level of functioning at start of rehabilitation. Although they might have an impact on the outcome of rehabilitation and on a person's well-being, we expected this to be similar for both groups. We also did not include a measure on cognitive impairment, which might be (more) present in DM1 and might have impacted on goal setting and conceptualizing well-being. However, in the capability care group, we did expect a higher capability well-being, as capability care is designed to identify and address broader aspects impacting on well-being. As a result, capability care may have changed a person's expectations or aspirations in a different way than usual care. Nevertheless, participants were blinded for which treatment (usual care or capability care) they

were receiving. We consider this blinding sufficiently successful; even though in the capability care group 21% of participants provided reasons that could be attributed to capability care. As a comparison, 25% in the capability care group had no suspicion of group allocation and 21% in this group incorrectly suspected that they received usual care. A forthcoming process analysis might provide more insight into how capability care might have changed the care provided and therefore the participants' well-being.

To conclude, the measures we have chosen might not have been able to capture a change in well-being as conceptualised in the capability approach. Currently, there is no capability well-being measure that is suitable for the context of personalized (rehabilitation) care. While methods for evaluating the impact of interventions on capabilities are still in development and challenges need to be addressed, recommendations and guidance for this development are available [42].

**Content of capability care.** Capability care was implemented at the outpatient rehabilitation clinic of the national Center of Expertise for FSHD and DM1, where persons with NMD are invited yearly for a day of 'analysis and advice'. The treatment advice formulated by the multidisciplinary rehabilitation team can include advice for the person with NMD themself, advice for a local healthcare professional (in most cases the physical therapist), or advice for further outpatient rehabilitation treatment (usually in a regional rehabilitation centre). Although the focus of the advice may have changed by providing capability care, the subsequent local or regional treatment was not provided by healthcare professionals trained in the capability approach. We do not know whether implementation of capability care in a rehabilitation treatment setting might have led to a larger difference. Furthermore, rehabilitation already shifted towards a broader focus on well-being using a biopsychosocial model: the World Health Organisation's International Classification of Disability, Functioning and Health (ICF) [43,44]. Healthcare in general is provided on a spectrum from purely medical (e.g., acute care) to broader aspects impacting on well-being. Rehabilitation for persons with NMD might already have a position on this spectrum closer to well-being and thereby implicitly include aspects of the capability approach. The question is whether the shift on this spectrum that capability care adds is sufficiently large to show a contrast with usual rehabilitation for persons with NMD.

**Implementation of capability care.** Another possible reason for the limited contrast between the two groups is an incomplete implementation of capability care. Several healthcare professionals experienced struggles with providing capability care during the implementation process. To support the healthcare team, additional training sessions were organised and additional feedback on their consultations and team meetings was provided. Whether or not this has led to a successful implementation will be investigated in a forthcoming process analysis.

### Effect of rehabilitation: changes over 6 months

The results showed a significant improvement at follow-up on both subscales of the COPM. However, these did not reach the level of clinically relevant improvement. Initially, we decided to use values of 1.4 on the COPM-P and 1.9 on the COPM-S as indication of clinically relevant improvement as perceived by the person [12,19]. However, recent discussions on the cut-off value for clinically relevant improvement concluded that there is insufficient evidence to determine a cut-off value [39]. It is also suggested that this value may vary depending on person characteristics, their occupational performance issues, and contextual factors [39]. Therefore, we cannot determine whether our results show a clinically relevant improvement over time. However, the observed improvements on the COPM, ranging from 0.8 to 1.1 six months after a visit to the multidisciplinary outpatient rehabilitation clinic, suggest that the day of 'analysis and advice' along with the subsequent interventions, contributed to positive changes in participants' occupational performance and satisfaction.

### Study design

We used a controlled before-after study design with two groups of participants, whereby participants were blinded for group allocation. Due to the outpatient clinic setup, it was not possible to blind the professionals, e.g., by training half of

the healthcare professionals and randomise participants. This might be considered a limitation; however, it is also a strength that the same healthcare professionals provided care in both groups before and after training in capability care.

In addition to the quantitative outcome measures reported in this article, we used several qualitative methods, such as audio-recording of consultations and multidisciplinary team meetings, access to the notes in the electronic health record, interviews with participants and focus groups with healthcare professionals. The use of this mixed-methods design may provide us with more in-depth understanding of the current results and the added value of the capability approach in rehabilitation for persons with NMD and will be published separately.

### Future directions

Future studies are needed on how to measure well-being change in a care setting such as rehabilitation, that focuses on achieving individualized goals to enhance a person's well-being. Additionally, further research in a rehabilitation treatment setting is needed, as we hypothesize that the contrast between groups will be larger in a treatment setting. Furthermore, the use of the capability approach is not limited to rehabilitation. Future studies in healthcare can contribute to the body of knowledge on using the capability approach in clinical intervention development.

### Conclusions

We found no difference on well-being of persons with NMD between participants receiving rehabilitation inspired by the capability approach and usual rehabilitation. There are several possible explanations for this; one of them being that the COPM might not be able to capture changes in broader aspects of well-being (i.e., capabilities and functionings that are not daily occupations). Another reason might be related to the extent that we were able to implement the capability approach in rehabilitation in the context of a single outpatient clinic visit with analysis and advice. A forthcoming process analysis will provide an in-depth understanding of these reasons.

### Supporting information

**S1 Table. Means and standard deviations on the USER-P for both groups at baseline and follow-up.** USER-P: Utrecht Scale for Evaluation of Rehabilitation Participation.
(DOCX)

**S2 Table. Means and standard deviations on the EQ-5D-5L for both groups at baseline and follow-up.** EQ5D-5L: EuroQol-5D-5L; VAS: visual analogue scale; LSS: level sum score.
(DOCX)

**S3 Table. Means and standard deviations on the ICECAP-A for both groups at baseline and follow-up.** ICECAP-A: ICEpop CAPability measure for Adults.
(DOCX)

**S4 Table. Means and standard deviations on the SF-36 for both groups at baseline and follow-up.** SF-36: Medical Outcome Study Short Form-36; PF: physical functioning; RP: role limitations due to physical health problems; RE: role limitations due to emotional health problems; VT: vitality; MH: mental health; SF: social functioning; BP: bodily pain; GH: general health.
(DOCX)

**S5 Table. Significant covariates for the ANCOVA on secondary outcome measures.** USER-P: Utrecht Scale for Evaluation of Rehabilitation Participation; EQ5D-5L: EuroQol-5D-5L; VAS: visual analogue scale; LSS: level sum score; ICECAP-A: ICEpop CAPability measure for Adults; SF-36: Medical Outcome Study Short Form-36; PF: physical

functioning; RP: role limitations due to physical health problems; RE: role limitations due to emotional health problems; VT: vitality; MH: mental health; SF: social functioning; BP: bodily pain; GH: general health.
(DOCX)

**S1 Protocol. Study protocol version 1.4 dd 31-03-21.**
(PDF)

**S1 Checklist. Trendstatement checklist ReCap-NMD.**
(DOC)

## Acknowledgments

We would like to thank our research assistants Nina de Bakker, Kim Willems and Jana Zajec for supporting with data collection and statistician Marianne Jonker for support with data analysis. We would also like to thank Ilse Karnebeek, nurse practitioner, for her involvement in participant recruitment. Finally, we would like to thank all health care professionals involved in this study by providing care (both usual and capability) to the participants. Several authors of this publication are members of the Radboudumc Neuromuscular Center (Radboud-NMD), Netherlands Neuromuscular Center (NL-NMD) and European Reference Network for rare neuromuscular diseases (EURO-NMD).

## Author contributions

**Conceptualization:** Eirlys J. Pijpers, Bart Bloemen, Baziel G. M. van Engelen, Gert Jan van der Wilt, Jan T. Groothuis, Edith H. C. Cup.

**Data curation:** Eirlys J. Pijpers, Bart Bloemen.

**Formal analysis:** Eirlys J. Pijpers.

**Funding acquisition:** Baziel G. M. van Engelen, Gert Jan van der Wilt, Jan T. Groothuis, Edith H. C. Cup.

**Investigation:** Eirlys J. Pijpers, Bart Bloemen.

**Methodology:** Eirlys J. Pijpers, Bart Bloemen, Wija J. Oortwijn, Baziel G. M. van Engelen, Gert Jan van der Wilt, Jan T. Groothuis, Edith H. C. Cup.

**Project administration:** Eirlys J. Pijpers, Bart Bloemen.

**Resources:** Baziel G. M. van Engelen, Jan T. Groothuis.

**Supervision:** Wija J. Oortwijn, Baziel G. M. van Engelen, Gert Jan van der Wilt, Jan T. Groothuis, Edith H. C. Cup.

**Visualization:** Eirlys J. Pijpers, Bart Bloemen.

**Writing – original draft:** Eirlys J. Pijpers.

**Writing – review & editing:** Eirlys J. Pijpers, Bart Bloemen, Wija J. Oortwijn, Baziel G. M. van Engelen, Gert Jan van der Wilt, Jan T. Groothuis, Edith H. C. Cup.

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
