## [Decision Letter · Decision Letter 0]

23 Oct 2024

Dear Dr. Bloemen,

Thank you for submitting your manuscript to PLOS ONE. After careful consideration, we feel that it has merit but does not fully meet PLOS ONE’s publication criteria as it currently stands. Therefore, we invite you to submit a revised version of the manuscript that addresses the points raised during the review process.

We kindly ask that you carefully assess the statistical feedback, as it contains important suggestions that may require detailed consideration and adjustments. Once addressed, your revised manuscript can be resubmitted for further evaluation.

Should you have any questions or require clarification, please feel free to reach out.

We look forward to receiving your revised manuscript.

Kind regards,

Giorgia Coratti

Guest Editor

PLOS ONE

2. In the online submission form you indicate that your data is not available for proprietary reasons and have provided a contact point for accessing this data. Please note that your current contact point is a co-author on this manuscript. According to our Data Policy, the contact point must not be an author on the manuscript and must be an institutional contact, ideally not an individual. Please revise your data statement to a non-author institutional point of contact, such as a data access or ethics committee, and send this to us via return email. Please also include contact information for the third party organization, and please include the full citation of where the data can be found.

Additional Editor Comments:

Dear Dr. Bloeman,

Please find attached the reviewers' comments for your review and response. We kindly ask that you carefully assess especially the statistical feedback, as it contains important suggestions that may require detailed consideration and adjustments. Once addressed, your revised manuscript can be resubmitted for further evaluation.

Should you have any questions or require clarification, please feel free to reach out.

Best regards,

Giorgia Coratti

Reviewers' comments:

Reviewer's Responses to Questions

**Comments to the Author**

1. Is the manuscript technically sound, and do the data support the conclusions?

Reviewer #1: Yes

Reviewer #2: Partly

Reviewer #3: Yes

2. Has the statistical analysis been performed appropriately and rigorously?

Reviewer #1: I Don't Know

Reviewer #2: No

Reviewer #3: Yes

3. Have the authors made all data underlying the findings in their manuscript fully available?

Reviewer #1: Yes

Reviewer #2: No

Reviewer #3: Yes

4. Is the manuscript presented in an intelligible fashion and written in standard English?

Reviewer #1: Yes

Reviewer #2: Yes

Reviewer #3: Yes

Reviewer #1: The authors developed an interesting and contemporary study which will support the SMA community. Multidisciplinary care and personnalized approaches are key to focus on patients' needs but indeed a personallized/individualized rehabilitation needs a personallized/individualized outcome measure like a GAS for example. Patient's well-being within rehabilitation and more broader within treatement in general, usually also depend on their expectations, which are very individual and usually depending on baseline fucnctional level. I did not see the mention of expectations within this study and I believe that this could also have an impact on the outcome. Patients subgrup analyzes according to functional level or similar baseline criteria (COPM and other scales) could be interesting as well.

In addition, the training of HCPs outside of the rehabilittion outpatient clinic seems to be a key point that is indeed missing in order to implement adequatly the concept.

Finally, not only a longer follow-up should be taken into consideration but also adding at least a second "day of advice" over time as mentioned in the conclusion.

Reviewer #2: The authors present results from a study evaluating capability care to usual care among persons with neuromuscular diseases. The authors found changes within groups on several outcomes, but no difference between the groups. The manuscript will be strengthened if the authors consider the following points:

1. If authors report one-sided p-values, they need to state the direction of the hypothesis and report the corresponding p-value, not pick the significant one-sided p-value to report. In particular, authors report a significant one-sided p-value for ICECAP-A, but then state it is in the direction opposite of what they hypothesized, in which case, the corresponding p-value for their actual hypothesis would not be significant. If authors had wanted to consider an improvement or deterioration, they should report the 2-sided p-value.

2. Authors need to further justify why they did not include covariates in the analyses for the secondary outcomes. Just because covariates were not significant for primary outcomes does not guarantee a lack of significance with the secondary outcomes.

3. Why are the reported degrees of freedom for COPM-P and COPM-S different (lines 247, 248) when Table 3 says the sample sizes are the same and the same covariates are included in the models?

4. Table 4 and Table 7 (in the appendix) are both for ICECAP-A, but the information in the Usual Care rows is different. It is not clear why that is and if authors have the data on the full 29 in the Usual Care group, why that isn't reported in the main paper.

Minor points:

1. Authors should state the number of participants and time between assessments in the abstract.

2. line 111: "is was determined" should be "it was determined"

3. In Table 2, authors should include the standard deviation for age

Reviewer #3: This is a well designed and written paper that contributes to the understanding of the impact of rehabilitation in NMD.

I would recommend to expand the explanation on the intervention provided. As it's currently presented the study, could not be replicated by others.

**Do you want your identity to be public for this peer review?** For information about this choice, including consent withdrawal, please see our Privacy Policy

Reviewer #1: No

Reviewer #2: No

Reviewer #3: **Yes: ** Robert Muni Lofra

---

## [Author Response · Author response to Decision Letter 1]

29 Jan 2025

Dear Editor,

We have submitted our revised manuscript. The responses to the reviewers can be found in the uploaded document. With this revision we have also updated the data availability statement as requested.

I hope that with the updated documents and data availability statement the revision process can proceed.

Kind regards,

Eirlys Pijpers

---

## [Decision Letter · Decision Letter 1]

28 Feb 2025

Effectiveness of the capability approach in rehabilitation for persons with neuromuscular diseases: a controlled before-after study

PLOS ONE

Dear Dr. Pijpers,

Thank you for submitting your manuscript to PLOS ONE. After careful consideration, we feel that it has merit but does not fully meet PLOS ONE’s publication criteria as it currently stands. Therefore, we invite you to submit a revised version of the manuscript that addresses the points raised during the review process.

We look forward to receiving your revised manuscript.

Kind regards,

Tai-Heng Chen, M.D., Ph.D.

Academic Editor

PLOS ONE

Journal Requirements:

Additional Editor Comments:

(1) Methods section is incomplete. How N=60 participants were divided into 2 groups? Is this a RCT or not?  I noticed ref 12 was cited for more details, but manuscript to me self-contained, it should include information that minimally required for understanding the method.  

(2)Statistical analysis section says  “For an intervention effect to be present, both the COPM-P and COPM-S needed to show a significant difference between groups (p < 0.05)”. Should also the difference be more than the minimum clinically significant difference (MCID) for it to be practically important?

(3)Results for the main outcome measure was given as Pvalues (ie, statistical significance). More useful info for readers is the size of the effect (ie, difference in means with CI). Difficult to assess practical importance of observed difference when confidence intervals not presented. This info should be already available in your current analysis, why not report them? Same for table3 where mean difference is given but not the CI for that. Presenting CIs allows readers to see the observed difference relative to MCID given in sample size calculation section.

(4)Under the secondary outcome measures in results section authors talk about “subscales” that was never mentioned in methods. What are these subscales? I didn’t find the answer even in the protocol submitted as supplementary material.

(5)Need a careful re-reading of the manuscript to eliminate inconsistancies. For example, EQ5D-5L-VAS is described as a measure in 0-10 scale, this is contradictory to the information in table5 which are out of this range.

Reviewers' comments:

Reviewer's Responses to Questions

**Comments to the Author**

Reviewer #1: (No Response)

Reviewer #2: All comments have been addressed

2. Is the manuscript technically sound, and do the data support the conclusions?

Reviewer #1: Partly

Reviewer #2: (No Response)

3. Has the statistical analysis been performed appropriately and rigorously?

Reviewer #1: Yes

Reviewer #2: (No Response)

4. Have the authors made all data underlying the findings in their manuscript fully available?

Reviewer #1: Yes

Reviewer #2: (No Response)

5. Is the manuscript presented in an intelligible fashion and written in standard English?

Reviewer #1: Yes

Reviewer #2: (No Response)

Reviewer #1: Thank you very much for this interesting work.

Please find some comments for your review:

Elaborating/mentioning the impact of even slight cognitive impairement (only severe cogntive impairement was an exlusion criteria) on choosing activities/tasks/functionings, setting goals and cocneptualizing well-being would be interesting as this might represent a limiting factor especially with your DM1 population

296: typo "them"

How have patients been blinded? if they did not see any difference in the care you were providing, they would know they were not in the tested group, hence they might have been disappointed, hence this would impact their well-being?

In conclusion, i think that finding/investigating new approaches/concepts/frameworks to enhance rehabilitation in NMD is really important, but to me here the real difference between the rehabilitation provided following multidisciplinary care team meetings and the capability care is not crystal clear.

Reviewer #2: (No Response)

**Do you want your identity to be public for this peer review?** For information about this choice, including consent withdrawal, please see our Privacy Policy

Reviewer #1: No

Reviewer #2: No

---

## [Author Response · Author response to Decision Letter 2]

1 Apr 2025

Dear editor and reviewers,

We thank you for your feedback and have adapted our manuscript. The response to your feedback can be found in our cover letter that we uploaded.

Kind regards,

Eirlys Pijpers

---

## [Decision Letter · Decision Letter 2]

15 Jun 2025

Dear Dr. Pijpers,

Thank you for submitting your manuscript to PLOS ONE. After careful consideration, we feel that it has merit but does not fully meet PLOS ONE’s publication criteria as it currently stands. Therefore, we invite you to submit a revised version of the manuscript that addresses the points raised during the review process.

The reviewers agree that the previous comments have been adequately addressed in the revised manuscript, but few additional minor revisions are still required.

We look forward to receiving your revised manuscript.

Kind regards,

Julie Dumonceaux

Academic Editor

PLOS ONE

Journal Requirements:

Reviewers' comments:

Reviewer's Responses to Questions

**Comments to the Author**

Reviewer #1: All comments have been addressed

Reviewer #4: All comments have been addressed

Reviewer #5: All comments have been addressed

2. Is the manuscript technically sound, and do the data support the conclusions?

Reviewer #1: Yes

Reviewer #4: Yes

Reviewer #5: Yes

3. Has the statistical analysis been performed appropriately and rigorously?

Reviewer #1: Yes

Reviewer #4: Yes

Reviewer #5: Yes

4. Have the authors made all data underlying the findings in their manuscript fully available?

Reviewer #1: Yes

Reviewer #4: Yes

Reviewer #5: Yes

5. Is the manuscript presented in an intelligible fashion and written in standard English?

Reviewer #1: Yes

Reviewer #4: Yes

Reviewer #5: Yes

Reviewer #1: Thank you for submitting this rvised version fo your manuscript. You explained in a clearer way the concept of capability within your study and highlight the study limitations.

Reviewer #4: Dear Editor,

Thank you for the opportunity to review the revised version (R2) of this manuscript. I have read the authors’ responses to the previous round of reviewer comments as well as the revised manuscript.

I appreciate the authors’ efforts to clarify their study rationale, methodological design, and interpretation of findings. The revisions have improved the overall transparency of the manuscript, particularly regarding participant blinding, the training process for the capability care team, and the interpretation of non-significant findings. The strengthened explanation of the theoretical link between functionings and the COPM, as well as the improved description of the outcome measures, are helpful.

Nevertheless, I have a few remaining comments:

Interpretation of negative findings

The manuscript now more clearly states that the study found no difference in outcomes between capability care and usual care. However, the discussion could benefit from further elaboration on what this equivalence might mean in clinical terms. For instance, does this imply that capability care is a viable alternative without additional harm or cost? Or should this outcome be seen as a limitation of the intervention as implemented?

Potential ceiling effects and sensitivity

Given that both groups improved equally on the COPM, I suggest including a short discussion on the possibility of ceiling effects or limited sensitivity of the COPM to detect differences in this specific patient population or care context.

Generalisability and context

The intervention is described in a highly specialised Dutch tertiary care context. A short reflection on the potential challenges of implementing “capability care” in other settings (e.g., primary care, other countries) would enhance the value of the discussion.

Language and clarity

A final language check would be beneficial to further improve fluency. For example, some sentences remain long and abstract in the Results and Discussion sections.

Overall, the manuscript reports on a relevant and novel rehabilitation approach in a population with high clinical needs. Despite the lack of superiority in outcomes, the work contributes to the emerging evidence base around person-centred and values-based care frameworks in rehabilitation.

I am in favor of publication after minor revisions addressing the above points.

Reviewer #5: 

First off, I have no objections or criticisms of the underlying scientific conduct of the study or the write up. In its own terms it is conducting research professionally and competently. Nor will I be copy editing, despite some irritating lapses (e.g. the ICF is both mistitled and mis-referenced), although I would recommend someone reviewing the paper in detail for slips like this.

My concern (and the reason for the ‘major’ rather than ‘minor’ revision recommendation), is that it is fundamentally confused about Sen’s capability theory. Or to put it another way the primary proxy outcomes measures (COPM-P, -S) or the secondary outcome measures (although USER-P is closest) are conceptually unlinked, i.e. they are not proxies of capability at all. Ultimately, the problem is that capability is not quality of life (that is acknowledged) nor is it ICF functioning (see, e.g. https://doi.org/10.1016/j.alter.2013.08.003 ), It is very different. in Sen’s account it is a measure of political equality (or level of development) based on opportunities.

The authors ignore in their choice of outcome measures that capacities can only be assessed in terms of the external, physical, social, political, economic resources that the person has access to and actually uses. This is not just a matter of a person’s intrinsic functioning ability (what they can do, after rehabilitation), nor is it a matter of their satisfaction with what they can do. It is a measure of what, in their actual circumstances, they can actually do. In ICF terminology it is a measure of performance not capacity. Unfortunately, teasing apart the impact of environmental supports, resources, etc from the actual performance has proven extremely different -- for both the ICF approach and capabilities. As a result, Sen’s account has been difficult to operationalize and use in practice. From Sen’s point of view, that is not a problem since his is a contribution to political theory more than direct application, although there is some literature suggesting that Nussbaum or Venkatapuram offer capability accounts of health that are more amenable to operationalization.

So, the foundations of this paper are problematic. I don’t see how that can be easily fixed since it foundational.

Perhaps, in fairness, if the paper acknowledged that their capability approach is not Sen’s (or the others in this area) but one inspired by the insights of the capability approach (or something else as modest), then readers won’t be mislead. And indeed that would be a minor revision.  

**Do you want your identity to be public for this peer review?** For information about this choice, including consent withdrawal, please see our Privacy Policy

Reviewer #1: No

Reviewer #4: No

Reviewer #5: No

---

## [Author Response · Author response to Decision Letter 3]

9 Jul 2025

Revision 3 (dd 09-07-2025):

Dear Julie Dumonceaux,

We have revised our manuscript and uploaded it in the system, including a cover letter responding to the reviewers.

Kind regards,

Eirlys Pijpers

corresponding author

---

## [Editor Report · Decision Letter 3]

1 Sep 2025

Effectiveness of the capability approach in rehabilitation for persons with neuromuscular diseases: a controlled before-after study

PONE-D-24-16662R3

Dear Dr. Pijpers,

We’re pleased to inform you that your manuscript has been judged scientifically suitable for publication and will be formally accepted for publication once it meets all outstanding technical requirements.

Kind regards,

Julie Dumonceaux

Academic Editor

PLOS ONE
---

## [Editor Report · Acceptance letter]

PONE-D-24-16662R3

PLOS ONE

Dear Dr. Pijpers,

I'm pleased to inform you that your manuscript has been deemed suitable for publication in PLOS ONE. Congratulations! Your manuscript is now being handed over to our production team.

Kind regards,

on behalf of

Dr. Julie Dumonceaux

Academic Editor

PLOS ONE